# Retrieval of Arctic Sea Ice Motion from FY-3D/MWRI Brightness Temperature Data

**Haihua Chen** [1] , **Kun Ni** [1,2] , **Jun Liu** [1] **and Lele Li** [1,*]

1 College of Marine Technology, Faculty of Information Science and Engineering, Ocean University of China, Qingdao 266100, China; chh7791@ouc.edu.cn (H.C.); nikun@stu.ouc.edu.cn (K.N.); junliu@stu.ouc.edu.cn (J.L.)
2 Aerospace Times FeiHong Technology Company Limited, Beijing 100094, China
* Correspondence: lilele@ouc.edu.cn; Tel.: +86-532-6678-1176

**Abstract:** Sea ice motion (SIM) has significant implications for sea–air interactions, thermohaline circulation, and the development of the Arctic passage. This research proposes an improved SIM retrieval method from Fengyun-3D's (FY-3D) microwave radiometer imager's (MWRI) brightness temperature ($T_b$) data based on the modified classical maximum cross-correlation (MCC) method and the multisource data merging method. This study utilized buoy data to establish the search area range, applied distinct thresholds across various Arctic regions, and merged the buoy data, reanalysis wind data, and SIM retrieved from FY-3D/MWRI $T_b$ data. In 2019, for the final Arctic SIM results retrieved from the MWRI 89 GHz and 36.5 GHz $T_b$ data, the root-mean-square error (RMSE) and the mean average error (MAE) in the east–west direction were 2.07 cm/s and 1.38 cm/s and those in the north–south direction were 1.96 cm/s and 1.15 cm/s, compared to the ice-tethered profiler (ITP) data. Compared with the daily average data of the National Snow and Ice Data Center (NSIDC), the RMSE and MAE of the SIM results obtained in this study were 0.74 cm/s and 0.93 cm/s in the east–west direction, and 0.56 cm/s and 0.72 cm/s in the north–south direction, respectively. The monthly average of the SIM retrieved from the MWRI $T_b$ data in this research also showed a good agreement with the monthly average of the NSIDC SIM product. The comparison showed that the MWRI $T_b$ data could be used to retrieve the Arctic SIM, and the Arctic SIM retrieval method presented in this paper was accurate and general.

**Keywords:** arctic sea ice motion (SIM); FY-3D/MWRI; brightness temperature; maximum cross-correlation





## 1. Introduction

The Arctic has a significant impact on Earth's climate. As a major cold source on the Earth, the Earth's climate change is amplified by 1.5 to 4.5 times in the Arctic region [1–3], which affects the Arctic circulation and energy flux and greatly enhances global climate and environmental changes. The Arctic climate has undergone significant transformations since the mid-20th century, and these rapid changes have driven alterations in the heat balance structure of the region, which have further contributed to climate change all around the world. Sea ice is an essential regulator and key factor influencing the Arctic climate [4,5], which can affect climate change, material balance, and sea level anomalies in the Arctic and globe. SIM is an important feature of sea ice and has an essential influence on sea ice change in the Arctic; SIM causes regional sea ice transport, and about 10% of the total Arctic sea ice is imported into the North Atlantic Ocean through the Fram Strait every year [6], which affects the sea ice mass balance; at the same time, the regional transport of sea ice also has a far-reaching influence on the thermohaline circulation conditions in the North Atlantic Ocean [4,7–10].

Since satellite remote sensing emerged at the end of the 20th century, it has been incorporated into Arctic sea ice observations and studies by many researchers. The Fengyun-3 (FY-3) series of satellites can provide observation data for meteorological forecasting

and environmental monitoring. A number of scholars have carried out studies on the Arctic environment based on the FY-3 satellite data, with numerous results [11–13]. Li et al. retrieved the Arctic sea ice concentration (SIC) and snow depth from the FY-3B/MWRI $T_b$ data in 2019 [11,12]. Chen et al. cross-calibrated the FY-3B/MWRI $T_b$ data with the Aqua/AMSR-E $T_b$ data in 2021 [13]. In 2022, Ni et al. retrieved SIM in the Beaufort Sea using the FY-3D/MWRI $T_b$ data [14].

The maximum correlation method was firstly used to retrieve SIM in the Arctic region by Ninnis et al. in 1986 [15]. Since then, the MCC method has been widely used by many researchers to retrieve SIM based on satellite data in the Arctic. In 1998, Kwok et al. retrieved the Arctic SIM from the Special Sensor Microwave Imager (SSM/I) $T_b$ data and validated the results using the motion of buoys and SIM retrieved from synthetic aperture radar (SAR) [16]. In 2000, Martin et al. used SSM/I 85.5 GHz $T_b$ data to retrieve the Arctic SIM and compared the results with the data of the buoys [17]. Numerous improvements to the MCC method have also been made by scholars. A bilinear interpolation was applied to the MCC approach by Lavergne et al. in the SIM retrieval [18], and Ezaty et al. used the Laplacian operator in their data preprocessing step and fused the results of different polarization inversions [19]. Liu et al. used a wavelet analysis to highlight sea ice features and improve SIM retrieval accuracy [20]. In 2017, a Laplacian computation of the Gaussian filter was used by Wang et al. to preprocess the $T_b$ data from the HY-2 satellite to retrieve SIM in the Arctic [21]. Regarding the sea ice retrieval method based on SAR data, Kwok et al. created a new method by combining template matching and feature tracking to retrieve SIM in Alaska [16]. In 2014, Komarov [22] used phase correlation and cross-correlation methods to track SIM from RADARSAT-2 ScanSAR images, with the HV channel tracking conditions being more reliable than the HH channel. Howell, Komarov et al. [23] used Sentinel-1A and ScanSAR to calculate the large-scale SIM. An open-source SIM retrieval method combining template matching and feature tracking was provided in 2017 [24], and it was used to retrieve SIM from the Sentinel-1 SAR data in parts of the Fram Strait. In 2022, Li et al. retrieved SIM using Sentinel-1 SAR data in parts of the Arctic with a feature tracking algorithm and corrected the results using a different vector filter [25]. Another important research field related to SIM is the validation of SIM data product quality. In 2013, Hwang et al. validated SIM products with different resolutions using ice-tethered profilers' (ITP) motion data [26]. In 2020, Shi et al. validated a SIM product using buoy data from the International Arctic Buoy Program (IABP) [27]. In 2021, Wang et al. verified the quality of SIM data products using buoy data from IABP and MOSAiC, the Multidisciplinary Drift Observatory for Arctic Climate Research [28].

Currently, no operational SIM data products based on FY-3D satellite data have been released. In this paper, we developed a method for the retrieval of SIM in the Arctic during 2019 using 89 GHz and 36.5 GHz data from FY-3D/MWRI $T_b$ data. Compared to our SIM retrieval method for the Beaufort Sea [14], we improved the algorithm, and the method studied in this paper is applicable to the entire Arctic region and avoids the underestimation of the SIM retrieval results. The paper is organized as follows: Section 2 presents the data materials and methodology; Section 3 provides the SIM results; Section 4 discusses the SIM results; Section 5 is devoted to the conclusions of this study.

## 2. Materials and Methods

### 2.1. Data

The datasets contained the FY-3D/MWRI $T_b$ data, sea ice concentration (SIC) product, IABP buoy data, ITP data, and NSIDC SIM product in the Arctic from January to December 2019. The FY-3D/MWRI 89 GHz and 36.5 GHz $T_b$ data obtained from the National Satellite Meteorological Center (NSMC) were used to retrieve the SIM. The sea ice concentration data from the Key Laboratory of Polar Oceanography and Global Ocean Change (POGOC) were used to distinguish water and sea ice. The IABP buoy data and NCEP/NCAR reanalysis wind data [29] were used for data assimilation. The ITP data from the Woods

Hole Oceanographic Institution (WHOI) and the SIM product from NSIDC [30] were used to validate the SIM retrieved in this research.

### 2.2. Data Preprocessing

First, we calculated the motion of the IABP buoys and ITP buoys based on their locations at different times. We subsequently mapped all the data used in this study into a polar stereographic grid with a spatial resolution of 12.5 km. In this coordinate system, the east and north directions are the positive directions, and the west and south directions are the negative directions. Third, the daily average $T_b$ data from FY-3D/MWRI were calculated, and the SIC data were used to distinguish seawater from sea ice. The $T_b$ data with sea ice concentrations higher than 15% were considered as sea ice, while $T_b$ data with sea ice concentrations lower than 15% were removed [31]. In addition, we used a Laplacian of Gaussian (LOG) filter to enhance the sea ice features in the images [21]. Figure 1 shows the 1 January 2019 $T_b$ data and the research area. The motion of the IABP buoy from 1 January 2019 to 4 January 2019 and the motion of the ITP buoy in 2019 are shown in Figure 2.

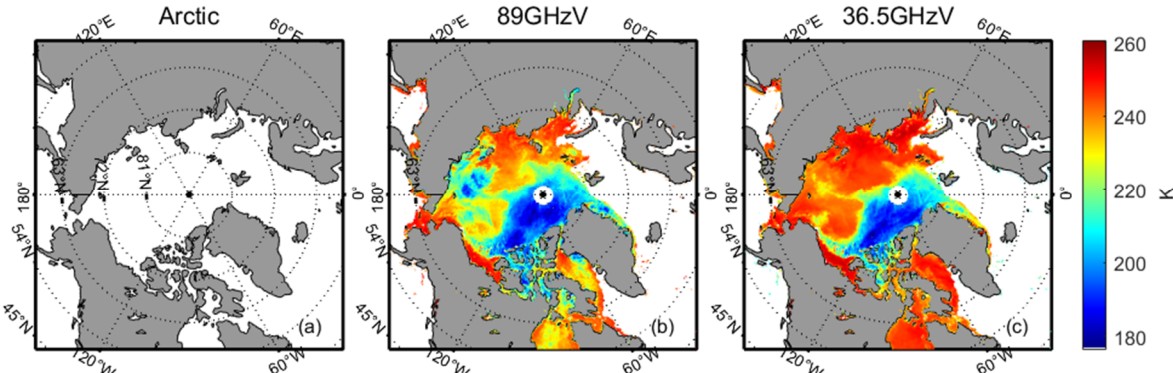

**Figure 1.** The Arctic research area and the 89 GHz and 36.5 GHz $T_b$ data on 1 January 2019.

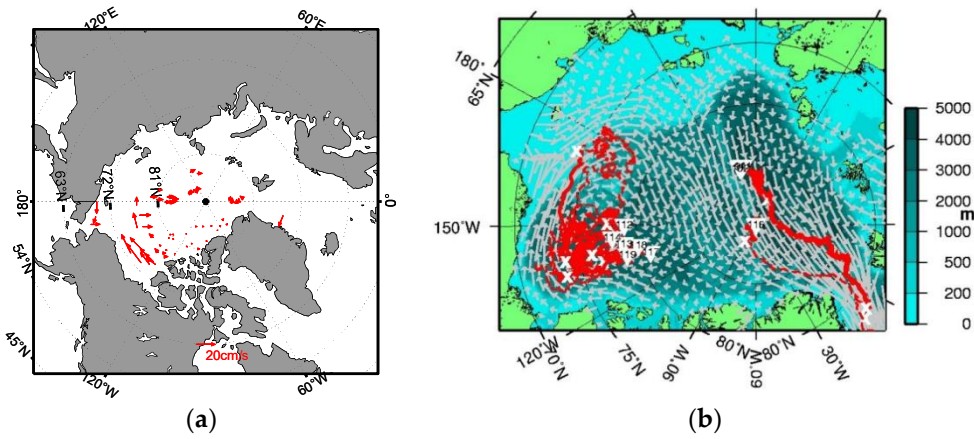

**Figure 2.** (**a**) IABP buoy data from 1 to 4 January 2019, (**b**) the motion of ITP buoys in 2019 (https://www2.whoi.edu/site/itp/, accessed on 23 July 2023). The triangle indicates the number of the buoy where it was deployed, the cross is the last position of the buoy, and the red is the drift track.

### 2.3. SIM Retrieval Method in the Arctic

The SIM retrieval method studied in this paper consisted of the following two parts: a modified MCC method and a multi-source data merging method based on the successive correction method (SCM) [14] data assimilation.

In this study, we applied the modified method to retrieve the SIM in the Arctic. Figure 3 shows the technical execution flowchart of this paper, which consisted of three parts: data preprocessing, SIM retrieval, and SIM validation.

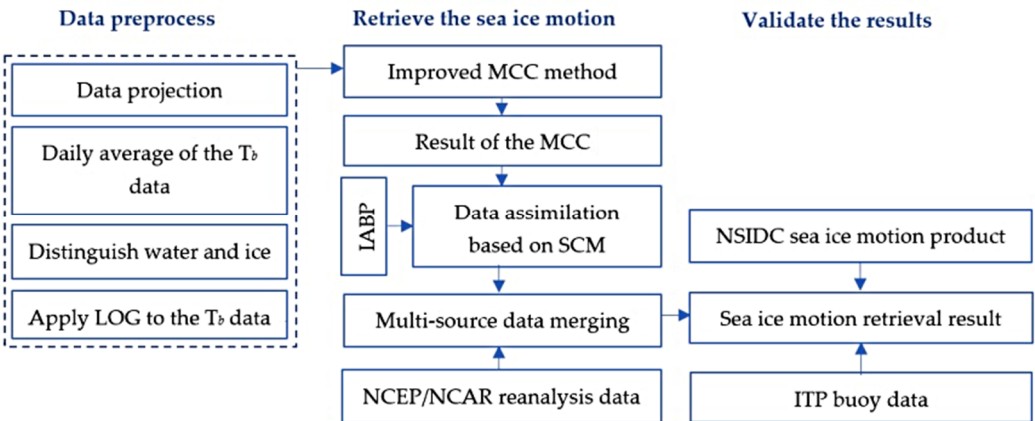

**Figure 3.** Technical execution in this study.

The MCC algorithm [14] is characterized by its computational simplicity and strong reliability, but it also suffers from quantization error and the effect of image noise. To solve these problems, the MCC algorithm was improved in this paper. The main improvements we made to the SIM retrieval method were as follows.

First, we identified the search range using the maximum of the IABP buoy data. According to the maximum value of the daily IABP buoy data, the maximum distance of the SIM for that day was calculated, which became the search range in the MCC method.

Second, for the setting of the other parameters in the modified MCC method, after many experiments, we set the size of the template at $7 \times 7$ and the time interval at 3 days. The template size contained enough sea ice features to maintain a high computation speed. The time interval also ensured that the scale of the SIM could be detected.

Third, considering the sea ice conditions in different Arctic regions, we set different thresholds for the MCC method in different regions. The threshold above $80^\circ$ N was set to 0.6 in area where sea ice changes dramatically and SIM is very rapid; the threshold in the region below $80^\circ$ N was set to 0.4, as the SIM is slower in that region.

Fourth, in view of the impact of other factors on the SIM retrieval, in this study, we developed a multisource data merging method that combines data from the wind, IABP buoys, and SIM derived from MWRI $T_b$ to obtain the final fused SIM result. The merging method was developed based on SCM data assimilation [14,32–34]. In this paper, we assimilated the above multisource data by the following weight average formula:

$$SIV = \frac{\sum_{i=1}^{n_w} w_w v_w + \sum_{i=1}^{n_b} w_b v_b + \sum_{i=1}^{n_{sat}} w_{sat} v_{sat}}{15} \tag{1}$$

where *SIV* is the velocity of the sea ice, and $w_w$, $w_b$, and $w_{sat}$ are the weights assigned to the wind velocity data, IABP buoys data, and the retrieved SIM from MWRI $T_b$ data, respectively. Drawing on the research by Tschudi et al. [31], the formula for calculating the weight $w$ is as follows:

$$w = Ce^{-\frac{d}{D}} \tag{2}$$

where $w$ is the weight and $C$ indicates the coefficient based on the data source. In this study, this coefficient was determined by the correlation coefficient between each data source and the buoy motion data (the true value of sea ice drift), which was used to obtain the final sea ice drift inversion results. d represents the distance between the raster data to be retrieved and the other surrounding data sources, $D$ is the influence radius of each type of data [31], and all kinds of data with a distance less than $D$ from the data point to be retrieved would have an influence on the current calculated data point.

The top 15 highest weighted data were used to conduct multisource data merging, as shown in Formula (1).

## 3. Results

### *3.1. Retrieval Results of Arctic SIM Based on Modified MCC Method*

3.1.1. SIM Inversion from $T_b$ Data with Different Polarization

We applied the modified MCC method to the FY-3D/MWRI $T_b$ data from 89 GHz and 36.5 GHz to retrieve the Arctic SIM. Figures 4 and 5 show the preliminary results of the SIM using the modified MCC method with the 89 GHz and 36.5 GHz $T_b$ data from January 1 to 4, 2019, respectively. The figures show that the SIM retrieved from $T_b$ data at the same frequency with different polarizations was roughly the same.

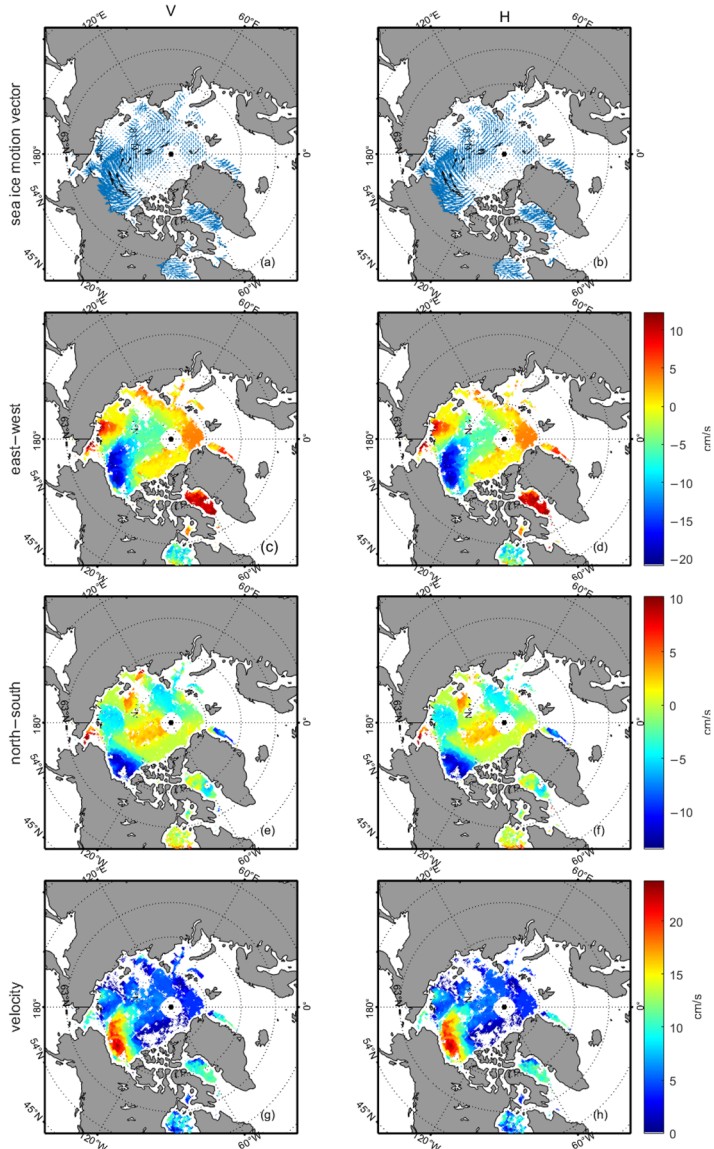

**Figure 4.** SIM retrieved from the 89 GHz FY-3D/MWRI $T_b$ data. The first and second columns are the SIM results obtained from V and H polarization $T_b$ data, respectively. (**a,b**) show the SIM vector, (**c,d**) show the east–west SIM velocity, (**e,f**) show the north–south SIM velocity, and (**g,h**) show the absolute values of the SIM velocities.

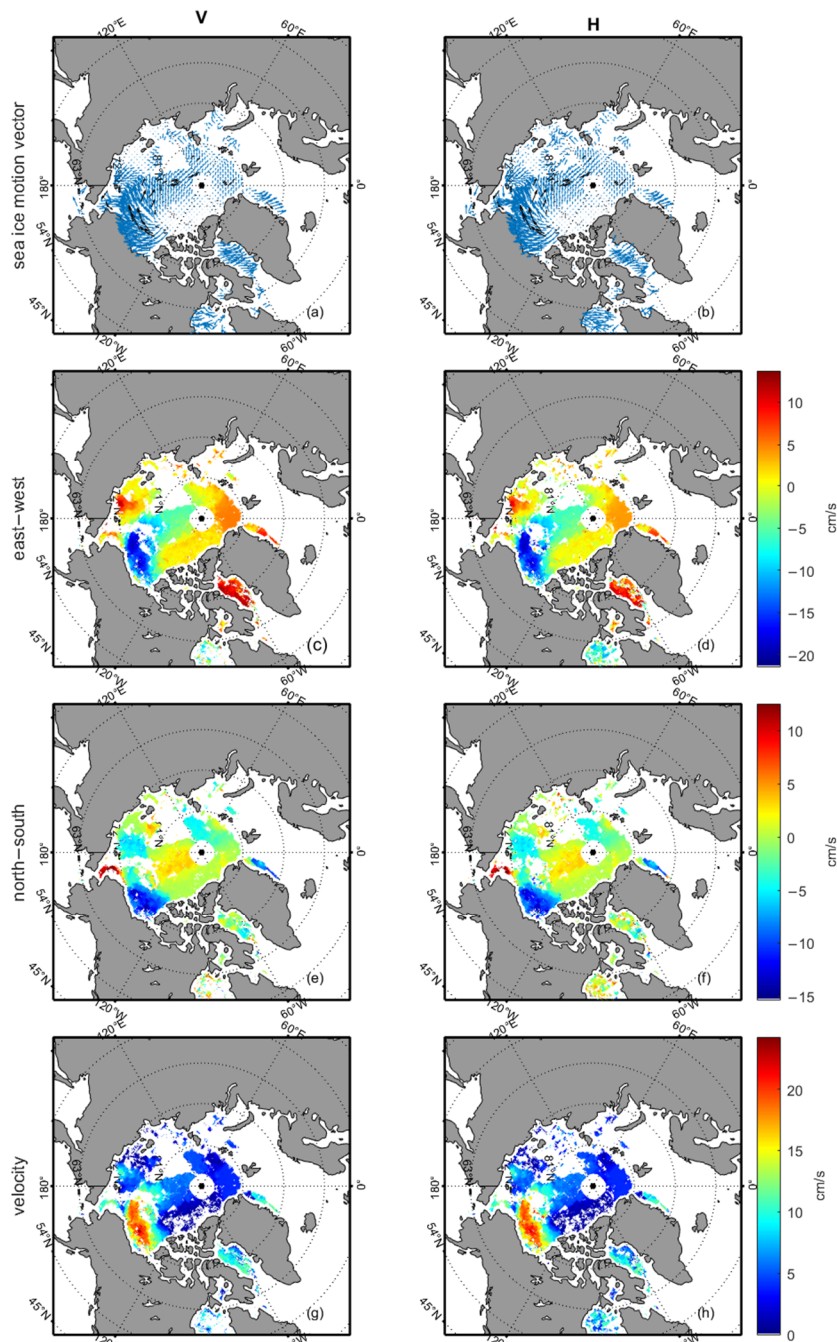

**Figure 5.** SIM retrieved from 36.5 GHz FY-3D/MWRI T$_b$ data. The first and second columns are the SIM results obtained from V and H polarization T$_b$ data, respectively. (**a**,**b**) show the SIM vector, (**c**,**d**) show the east–west SIM velocity, (**e**,**f**) show the north–south SIM velocity, and (**g**,**h**) show the absolute values of the SIM velocities.

We matched the retrieved SIM with the IABP buoy data and calculated the root-mean-square error (RMSE) and mean absolute error (MAE) between them. In this study, the MAE parameter is denoted by δ and the RMSE is denoted by σ.

Table 1 shows the number of matching points for different seasons [4], with June to September defined as summer and the other months as winter in this study.

**Table 1.** Statistics of the number of matching points.

| | 36.5 GHz | | 89 GHz | |
|---|---|---|---|---|
| | H | V | H | V |
| Winter | 9764 | 10,607 | 8472 | 9098 |
| Summer | 617 | 1032 | 468 | 590 |
| Total number | 10,381 | 11,639 | 8940 | 9688 |

Tables 2 and 3 show the RMSE and MAE of the retrieved SIM from the MWRI $T_b$ data compared to the IABP buoy motion data.

**Table 2.** Comparison between the SIM results from the 89 GHz $T_b$ data and IABP buoy motion data.

| | | H | | V | |
|---|---|---|---|---|---|
| | | East–West $\delta/\sigma$ (cm/s) | North–South $\delta/\sigma$ (cm/s) | East–West $\delta/\sigma$ (cm/s) | North–South $\delta/\sigma$ (cm/s) |
| Winter | January | 2.42/3.33 | 2.58/3.83 | 2.43/3.36 | 2.62/3.83 |
| | February | 3.95/4.85 | 3.39/4.33 | 3.88/4.70 | 3.30/4.29 |
| | March | 2.35/3.43 | 2.52/3.63 | 2.34/3.39 | 2.52/3.72 |
| | April | 3.22/4.39 | 3.28/4.41 | 3.22/4.46 | 3.13/4.24 |
| | May | 3.51/4.91 | 4.56/6.47 | 3.49/4.67 | 4.09/5.74 |
| | June | 4.73/6.40 | 5.63/8.36 | 4.74/6.69 | 4.64/7.24 |
| Summer | July | 6.27/8.78 | 5.83/8.77 | 5.58/7.72 | 7.05/10.72 |
| | August | 7.3/10.05 | 7.38/11.1 | 6.72/8.89 | 7.53/10.23 |
| | September | 5.81/7.54 | 5.00/7.55 | 5.08/6.91 | 4.60/6.87 |
| | October | 3.51/4.76 | 5.17/6.57 | 3.43/4.57 | 5.13/6.55 |
| Winter | November | 4.92/6.25 | 5.21/6.68 | 4.84/6.17 | 5.08/6.44 |
| | December | 4.33/5.77 | 4.57/6.01 | 4.28/5.73 | 4.55/5.91 |
| | January–December | 3.70/5.07 | 4.06/5.62 | 3.65/4.98 | 3.97/5.50 |

**Table 3.** Comparison between the SIM results from the 36.5 GHz $T_b$ data and IABP buoy motion data.

| | | H | | V | |
|---|---|---|---|---|---|
| | | East–West $\delta/\sigma$ (cm/s) | North–South $\delta/\sigma$ (cm/s) | East–West $\delta/\sigma$ (cm/s) | North–South $\delta/\sigma$ (cm/s) |
| Winter | January | 2.50/3.51 | 2.50/3.66 | 2.50/3.50 | 2.56/3.75 |
| | February | 4.12/5.09 | 3.24/4.22 | 4.11/5.04 | 3.28/4.33 |
| | March | 2.41/3.51 | 2.60/4.05 | 2.36/3.49 | 2.45/3.51 |
| | April | 3.10/4.27 | 2.88/4.05 | 3.12/4.25 | 3.00/4.23 |
| | May | 2.95/3.99 | 3.41/4.78 | 3.16/4.27 | 3.56/4.99 |
| | June | 3.88/5.14 | 4.28/5.70 | 4.77/6.65 | 4.85/7.44 |
| Summer | July | 3.82/5.33 | 3.94/6.05 | 4.69/6.47 | 5.01/7.02 |
| | August | 5.19/6.65 | 5.74/8.37 | 5.83/7.94 | 5.46/7.88 |
| | September | 4.61/6.37 | 4.28/6.34 | 4.40/5.82 | 4.15/6.03 |
| | October | 3.32/4.46 | 4.83/6.24 | 3.46/4.58 | 4.73/6.10 |
| Winter | November | 4.93/6.31 | 4.81/6.16 | 4.85/6.12 | 4.83/6.20 |
| | December | 4.01/5.39 | 4.41/5.89 | 4.34/5.89 | 4.47/5.98 |
| | January–December | 3.59/4.89 | 3.73/5.15 | 3.69/5.01 | 3.81/5.28 |

In comparison with the IABP buoy data, Tables 2 and 3 show that the RMSE and the MAE of the SIM of the 89 GHz $T_b$ data for each month of 2019 ranged from 3.33 to 11.1 cm/s and 2.34 to 7.53 cm/s, and the RMSE and MAE of the SIM of the 36.5 GHz $T_b$ data ranged from 3.49 to 8.37 cm/s and 2.36 to 5.83 cm/s, respectively; for the 2019 full-year statistical analysis results, the RMSE and the MAE of the SIM of the 89 GHz $T_b$ data ranged from 4.98 to 5.62 cm/s and 3.65 to 4.06 cm/s, and the RMSE and the MAE of the SIM of the 36.5 GHz $T_b$ data ranged from 4.89 to 5.28 cm/s and 3.81 to 3.59 cm/s. Comparing the

above results, it can be seen that the SIM inversion results from the 36.5 GHz T$_b$ data were slightly better than those based on the 89 GHz T$_b$ data.

Tables 1–3 show that the number of matching points in winter was much larger than that in summer. During the summer, sea ice melted rapidly, the status of the Arctic sea ice changed dramatically, and atmospheric conditions in the summer were more complex than in the winter.

As a result, retrieval results for summer SIM had larger errors and fewer matching points. In addition, the T$_b$ data at 89 GHz were more susceptible to atmospheric conditions than the data at 36.5 GHz; thus, the error of the SIM retrieved from the 89 GHz T$_b$ data was larger than that of the 36.5 GHz T$_b$ data.

3.1.2. Merging SIM from FY-3D/MWRI T$_b$ Data at Different Frequencies

Based on the statistics of the errors shown in Tables 2 and 3, we merged the SIMs retrieved from different frequencies and polarization data using the follow equation:

For the SIM retrieved from data with different polarizations,

$$SIV\_arctic \begin{cases} \frac{SIV_H+SIV_V}{2}(SIV_H \neq nan, SIV_V \neq nan) \\ SIV_H(SIV_H \neq nan, SIV_V = nan) \\ SIV_V(SIV_V \neq nan, SIV_H = nan) \end{cases} \tag{3}$$

For the SIM retrieved based on different frequencies,

$$SIV\_arctic \begin{cases} SIV_{89}(SIV_{36.5} \neq nan, SIV_{89} \neq nan) \\ SIV_{36.5}(SIV_{36.5} \neq nan, SIV_{89} = nan) \\ SIV_{89}(SIV_{89} \neq nan, SIV_{36.5} = nan) \end{cases} \tag{4}$$

*SIV*_arctic is the obtained sea ice velocity in the Arctic area, and the subscripts *H* and *V* in Formula (3) represent the horizontal polarization and vertical polarization, respectively, while the subscripts 89 and 36.5 in Formula (4) represent the 89 GHz and 36.5 GHz T$_b$ data, respectively. As shown in the above formulas, we calculated the average of the SIM retrieved from the different polarizations of the T$_b$ data and then supplemented the SIM values from the 89 GHz T$_b$ data with the SIM results from the 36.5 GHz data.

As an example, Figure 6 shows the Arctic SIM after merging the SIM retrieved from different frequencies and polarization MWRI T$_b$ data on 1–4 January 2019. Comparing Figures 5 and 6, we can see that the completeness of the SIM was greatly improved.

The distribution of SIM matching points retrieved from the IABP buoy data and the FY-3D/MWRI T$_b$ data is shown in Figure 7. Figure 7a–d are the distribution of the matching points in summer (Figure 7a–b) and winter (Figure 7c–d), respectively. The number of matching points in winter was significantly larger than that in summer. As shown in Figure 7, compared to the IABP buoy, the RMSE and MAE were larger in summer than in winter.

Tables 2–5 show that the accuracy of the SIM retrieved from different frequencies and polarizations T$_b$ data changed very little after merging. However, there was a significant increase in the number of matching points, as can be seen in Table 5.

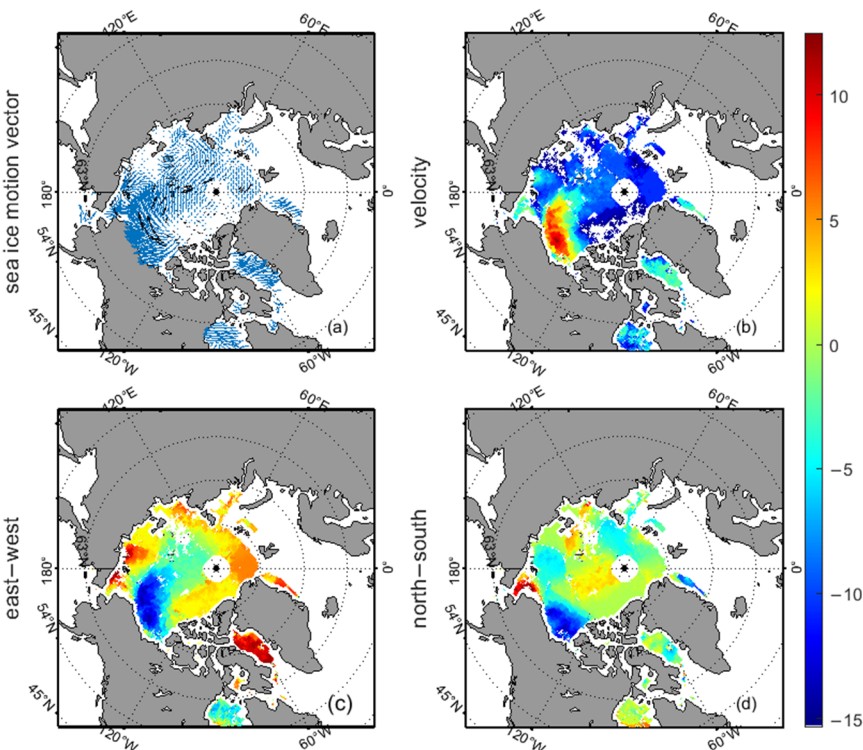

**Figure 6.** Merging results of the SIM retrieved from T$_b$ data at different frequencies and polarizations between January 1 and 4, 2019. (**a**) is the SIM vector, (**b**) is the SIM velocity, (**c**) is the SIM velocity in the east–west direction, (**d**) is the SIM velocity in the north–south direction.

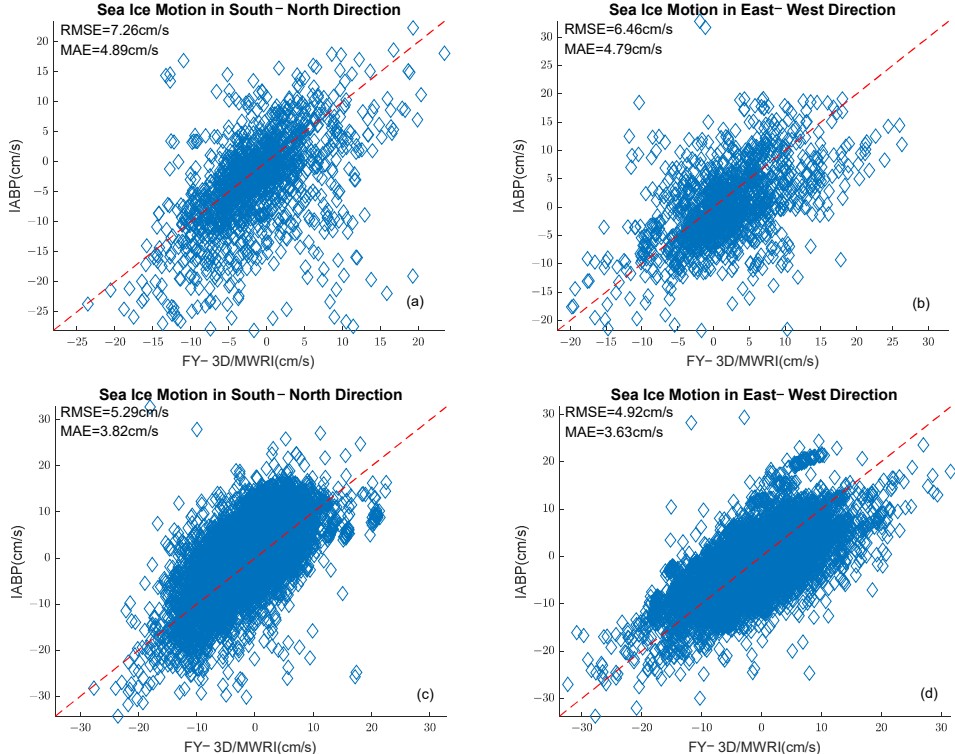

**Figure 7.** The distribution of matching points between the merging results of the SIM retrieved from the FY-3D/MWRI T$_b$ data at different frequencies and polarizations and the IABP buoy data in summer (**a**,**b**) and winter (**c**,**d**).

**Table 4.** Statistics of the RMSE and MAE between the merging SIM results and the IABP buoy data for each month of 2019.

|  |  | East–West $\delta/\sigma$ (cm/s) | North–South $\delta/\sigma$ (cm/s) |
|---|---|---|---|
| Winter | January | 2.56/3.60 | 2.62/3.84 |
|  | February | 4.12/5.10 | 3.27/4.36 |
|  | March | 2.41/3.55 | 2.56/3.97 |
|  | April | 3.26/4.47 | 3.26/4.34 |
|  | May | 3.28/4.42 | 3.74/5.28 |
| Summer | June | 4.70/6.31 | 5.18/7.84 |
|  | July | 4.64/6.53 | 4.96/7.39 |
|  | August | 5.82/7.84 | 6.15/8.90 |
|  | September | 4.55/5.97 | 4.29/6.20 |
|  | October | 3.53/4.74 | 4.72/6.13 |
| Winter | November | 4.79/6.06 | 4.92/6.36 |
|  | December | 4.39/5.92 | 4.61/6.15 |
| 2019 | January–December | 3.76/5.11 | 3.94/5.54 |

**Table 5.** Matching numbers of the SIM.

|  | Winter | Summer | Total Number |
|---|---|---|---|
| Number of matching points | 11,595 | 1454 | 13,049 |

### 3.2. Multisource Data Merging

For the multisource data merging of SIM in the Arctic, according to Formulas (1) and (2), we used the SCM method to assimilate the SIM retrieved from the MWRI $T_b$ data, IABP buoy motion data, and wind data to obtain the final SIM retrieval results.

Figures 8–11 show the distribution of the monthly average of the SIM retrieved from the FY-3D/MWRI $T_b$ data and the monthly average of the NSIDC SIM products for December, January, July, and August 2019, where December and January represent the typical winter months, while July and August represent the summer months in the Arctic. In each figure, the first row shows the SIM vector, the second row shows the SIM velocity in the east–west direction, the third row shows the SIM velocity in the north–south direction, and the fourth row shows the absolute value of the SIM velocity. The first column is the SIM retrieved from the FY-3D/MWRI $T_b$ data, and the second column is the NSIDC SIM product.

Figures 12 and 13 show the SIM in summer and winter, respectively. From Figures 8–13, we can see that the two distinctive features of the Arctic region, namely, the anti-cyclonic Beaufort vortex and the pole-penetrating drift, are reflected well. However, the agreement of the SIM retrieved from FY-3D/MWRI $T_b$ data in winter with the NSIDC SIM product was better than that of the SIM retrieved from FY-3D/MWRI $T_b$ data in summer.

Table 6 shows the statistics of the errors between the monthly average of the SIM retrieved from the FY-3D/MWRI $T_b$ data and the NSIDC SIM product.

**Table 6.** Comparison of the SIM retrieved from the MWRI $T_b$ data with NSIDC SIM products (AE is the average error).

|  | East–West (AE/RMSE(cm/s)) | North–South (AE/RMSE(cm/s)) | Sea Ice Velocity (AE/RMSE(cm/s)) |
|---|---|---|---|
| January | 0.26/0.88 | 0.24/0.76 | −0.94/0.89 |
| February | 1.47/0.99 | −0.13/0.85 | −0.88/0.99 |
| March | 0.66/0.95 | 0.18/0.89 | −0.91/0.95 |
| April | −0.63/0.79 | 0.26/0.83 | −0.33/0.79 |
| May | −1.04/0.68 | 0.39/0.72 | 0.32/0.69 |
| June | −0.29/0.58 | −0.30/0.56 | 0.35/0.58 |
| July | −0.38/0.49 | −0.08/0.50 | 0.51/0.49 |
| August | 0.19/0.40 | 0.27/0.38 | 0.49/0.40 |
| September | −0.19/0.41 | 0.75/0.45 | 0.36/0.41 |
| October | −0.38/0.81 | −0.90/0.90 | 0.50/0.81 |

**Table 6.** *Cont.*

|  | East–West (AE/RMSE(cm/s)) | North–South (AE/RMSE(cm/s)) | Sea Ice Velocity (AE/RMSE(cm/s)) |
|---|---|---|---|
| November | 0.044/0.76 | −0.02/0.66 | 0.02/0.76 |
| December | −0.35/0.69 | 0.33/0.70 | −0.36/0.69 |
| winter | 0.29/0.83 | 0.27/0.79 | −0.76/0.77 |
| summer | −0.20/0.47 | −0.03/0.45 | 0.36/0.49 |

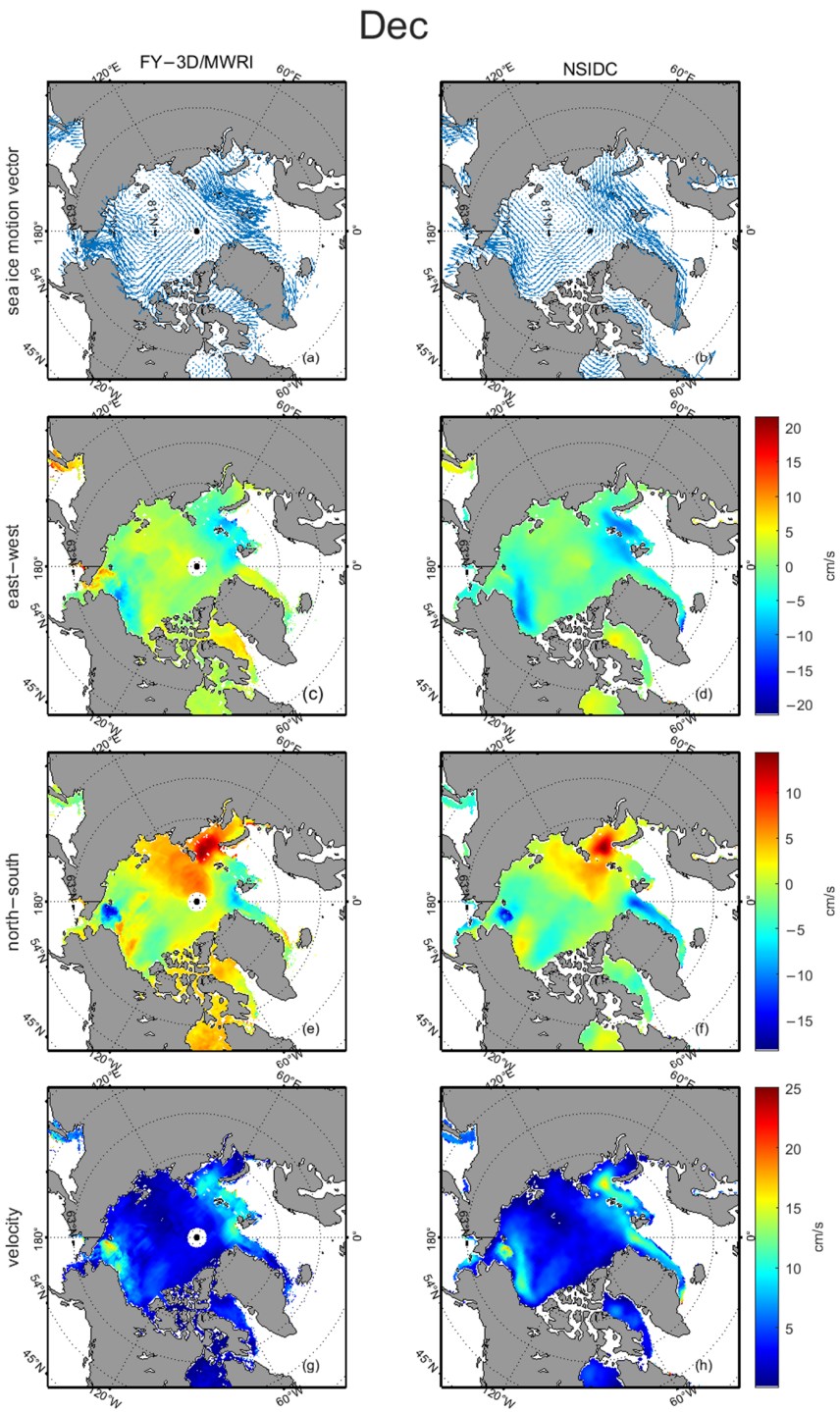

**Figure 8.** SIM in December 2019. The first and second columns are SIM retrieved from the FY-3D/MWRI Tb data and the monthly average of the NSIDC SIM products, respectively. (**a**,**b**) show the SIM vector, (**c**,**d**) show the east–west SIM velocity, (**e**,**f**) show the north–south SIM velocity, and (**g**,**h**) show the absolute values of the SIM velocities.

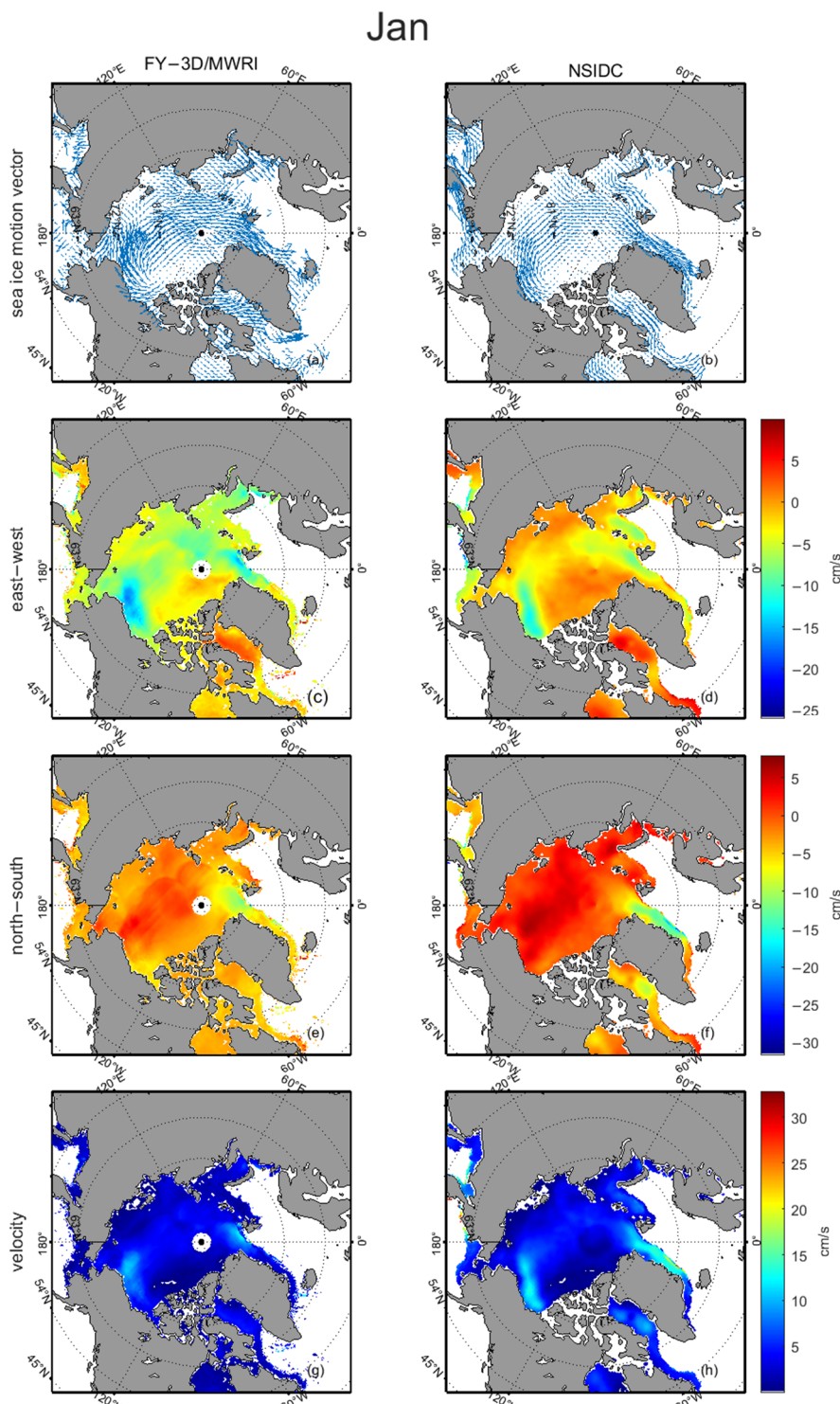

**Figure 9.** SIM in January 2019. The first and second columns are SIM retrieved from the FY-3D/MWRI Tb data and the monthly average of the NSIDC SIM products, respectively. (**a**,**b**) show the SIM vector, (**c**,**d**) show the east–west SIM velocity, (**e**,**f**) show the north–south SIM velocity, and (**g**,**h**) show the absolute values of the SIM velocities.

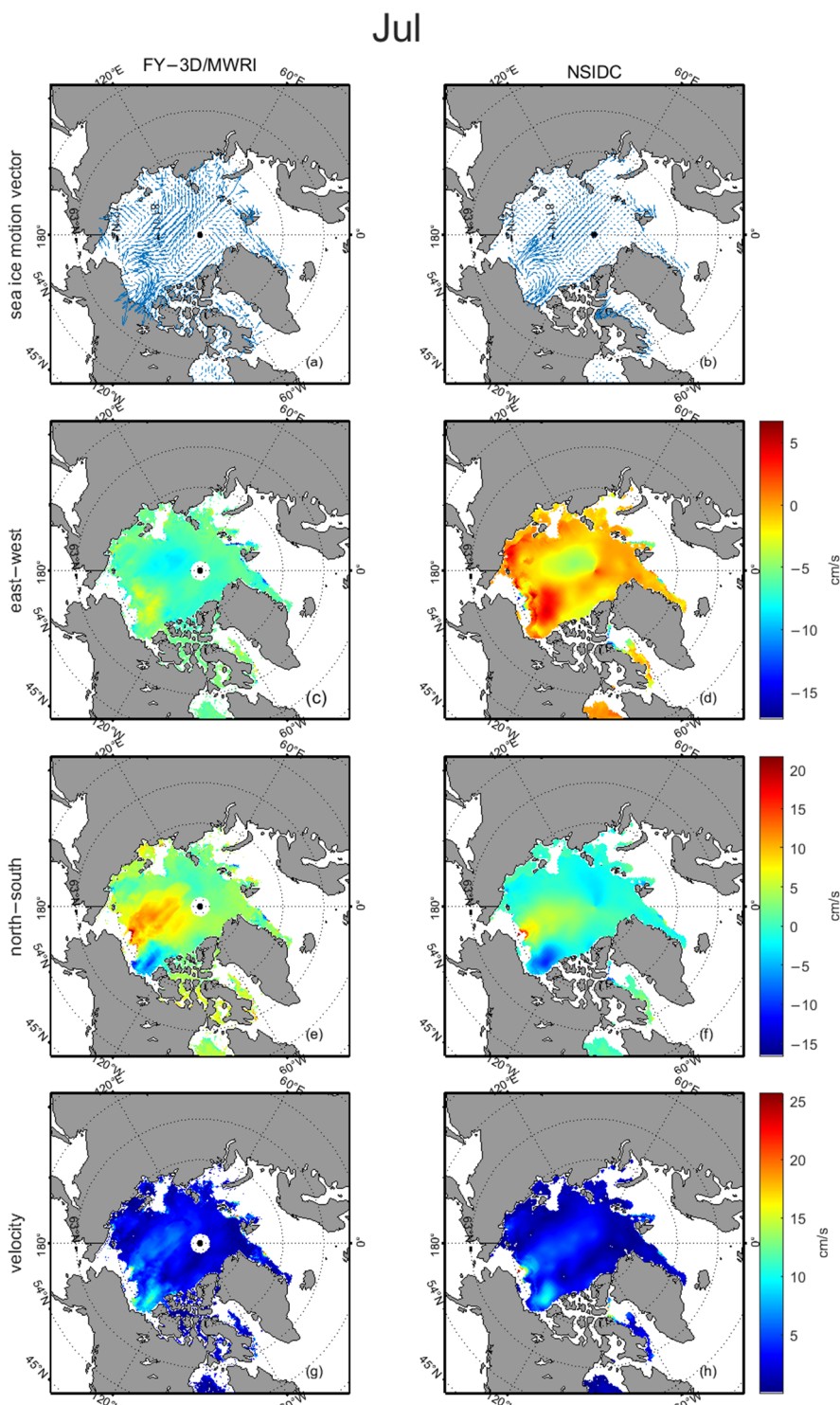

**Figure 10.** SIM in July 2019. The first and second columns are SIM retrieved from the FY-3D/MWRI Tb data and the monthly average of the NSIDC SIM products, respectively. (**a**,**b**) show the SIM vector, (**c**,**d**) show the east–west SIM velocity, (**e**,**f**) show the north–south SIM velocity, and (**g**,**h**) show the absolute values of the SIM velocities.

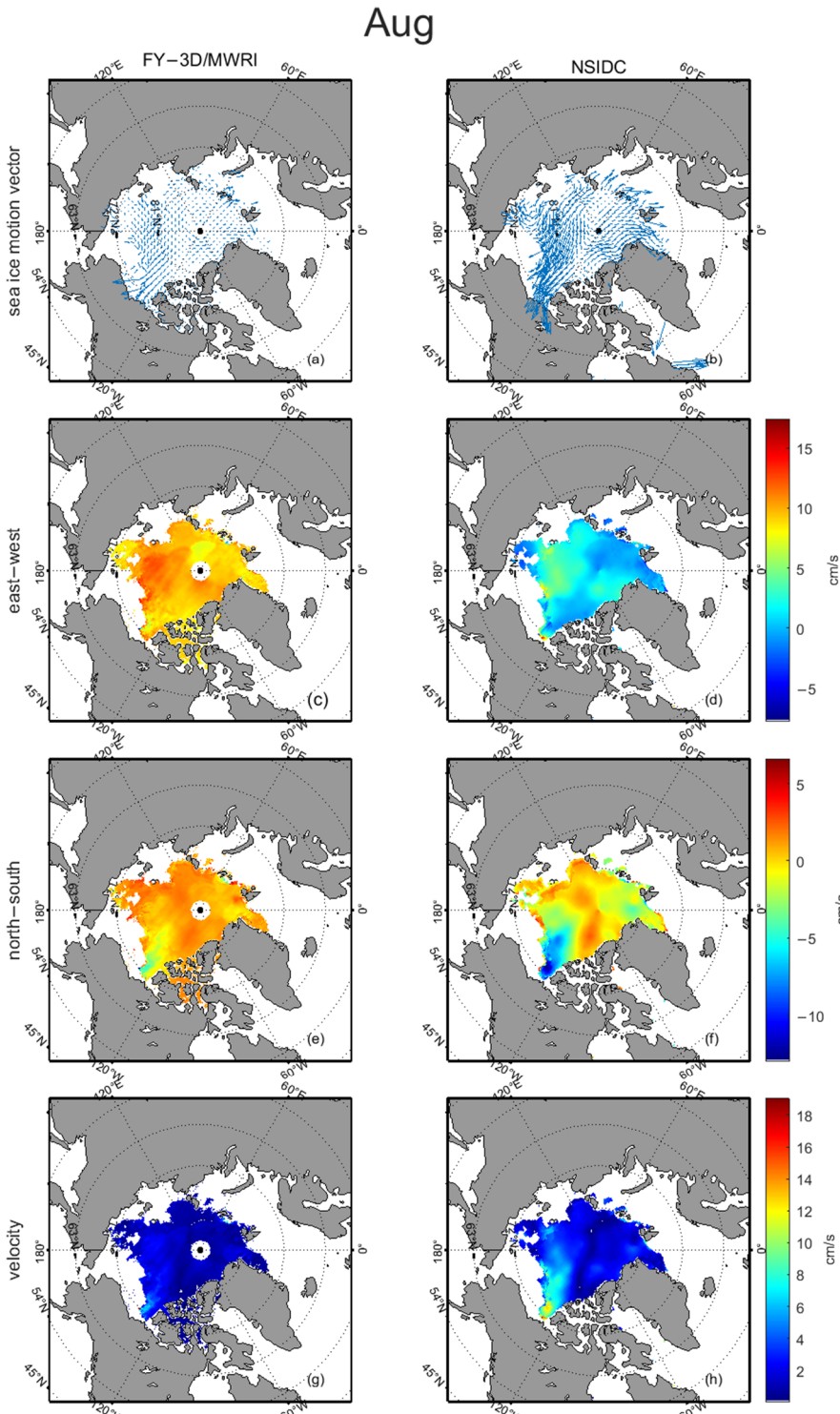

**Figure 11.** SIM in August 2019. The first and second columns are SIM retrieved from the FY-3D/MWRI Tb data and the monthly average of the NSIDC SIM products, respectively. (**a**,**b**) show the SIM vector, (**c**,**d**) show the east–west SIM velocity, (**e**,**f**) show the north–south SIM velocity, and (**g**,**h**) show the absolute values of the SIM velocities.

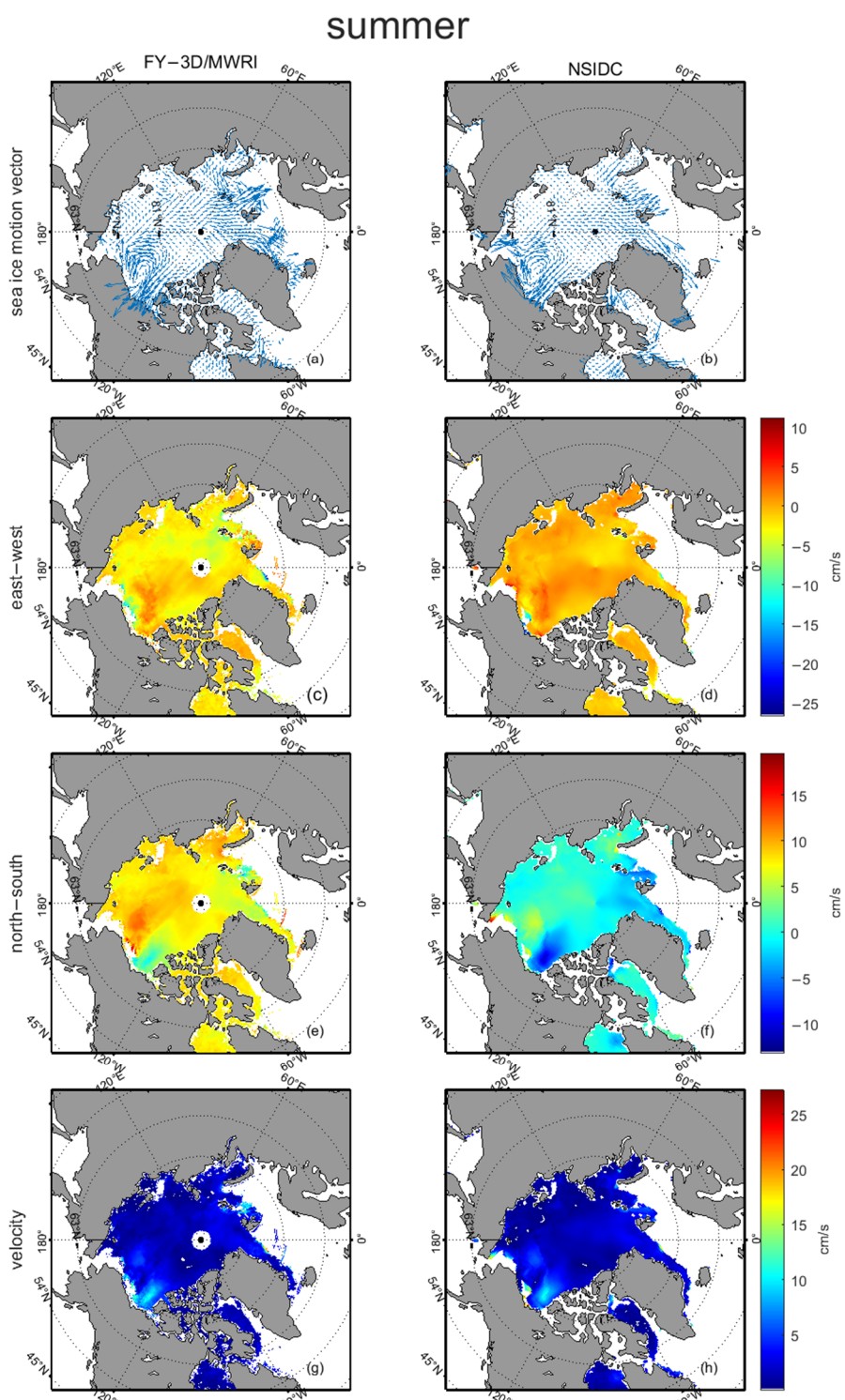

**Figure 12.** SIM in summer 2019. The first and second columns are SIM retrieved from the FY-3D/MWRI Tb data and the monthly average of the NSIDC SIM products, respectively. (**a**,**b**) show the SIM vector, (**c**,**d**) show the east–west SIM velocity, (**e**,**f**) show the north–south SIM velocity, and (**g**,**h**) show the absolute values of the SIM velocities.

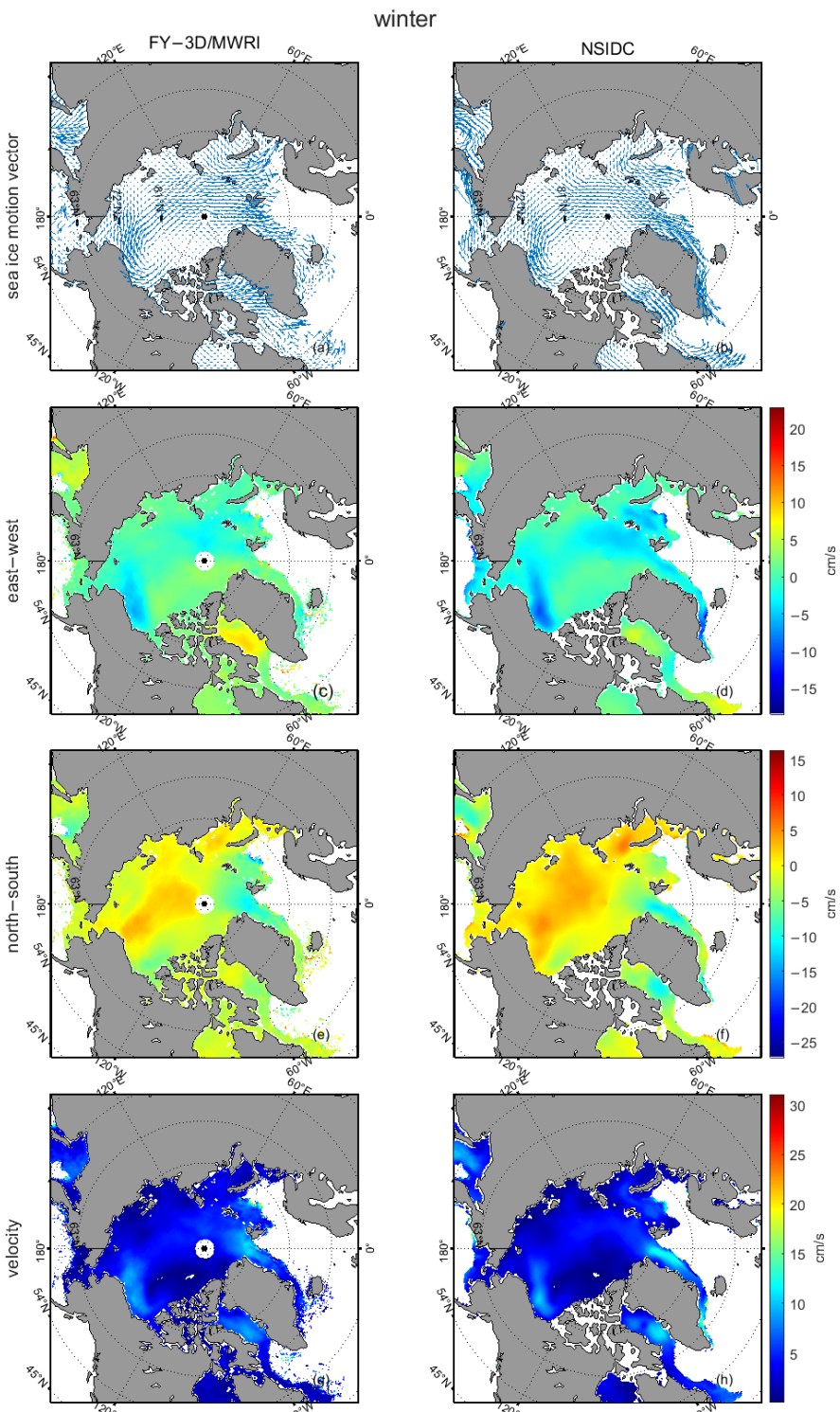

**Figure 13.** SIM in winter 2019. The first and second columns are SIM retrieved from the FY-3D/MWRI Tb data and the monthly average of the NSIDC SIM products, respectively. (**a,b**) show the SIM vector, (**c,d**) show the east–west SIM velocity, (**e,f**) show the north–south SIM velocity, and (**g,h**) show the absolute values of the SIM velocities.

As seen in Table 6, the SIM results retrieved from the $T_b$ data in the east–west direction had larger average errors in September (0.75 cm/s) and October (–0.90 cm/s). The errors indicated that the eastward SIM retrieved from the $T_b$ data for September was overestimated, and the westward SIM for October was overestimated.

In the north–south direction, the SIM had larger average errors in February (1.47 cm/s) and May (–1.04 cm/s), which shows that the northward SIM was overestimated in February, and the southward SIM was overestimated in May. For the velocity value of SIM, the average errors of the SIM retrieved from the $T_b$ data in January (–0.94 cm/s) and March (–0.91 cm/s) were larger than those of the other months. The errors indicated that SIM was underestimated in January and March.

In Table 6, the average errors of the SIM in winter were 0.29 cm/s and 0.27 cm/s in the north–south and east–west directions, respectively. The comparison shows that the SIM in winter was overestimated in the northward and eastward directions. The average errors of the SIM in summer were –0.20 cm/s in the north–south direction and –0.03 cm/s in the east–west direction, indicating that the SIM was overestimated in the south and west directions. In general, the performance of the SIM inversion results was greatly improved by the data assimilation.

## 4. Discussion

In this study, we evaluated the accuracy of the SIM retrieved from the FY-3D/MWRI $T_b$ data by calculating the motion of the ITP buoys as validation data in the Arctic region. The NSIDC SIM product was also used for the comparison data for the SIM retrieved in this paper.

Figure 14 shows a comparison of the daily average SIM among the retrieved SIM result after data fusion and assimilation in this paper, the NSIDC SIM, and the IABP buoy data in 2019. As can be seen in Figure 14, the daily average of the SIM retrieved from the $T_b$ data was in good agreement with the NSIDC SIM product.

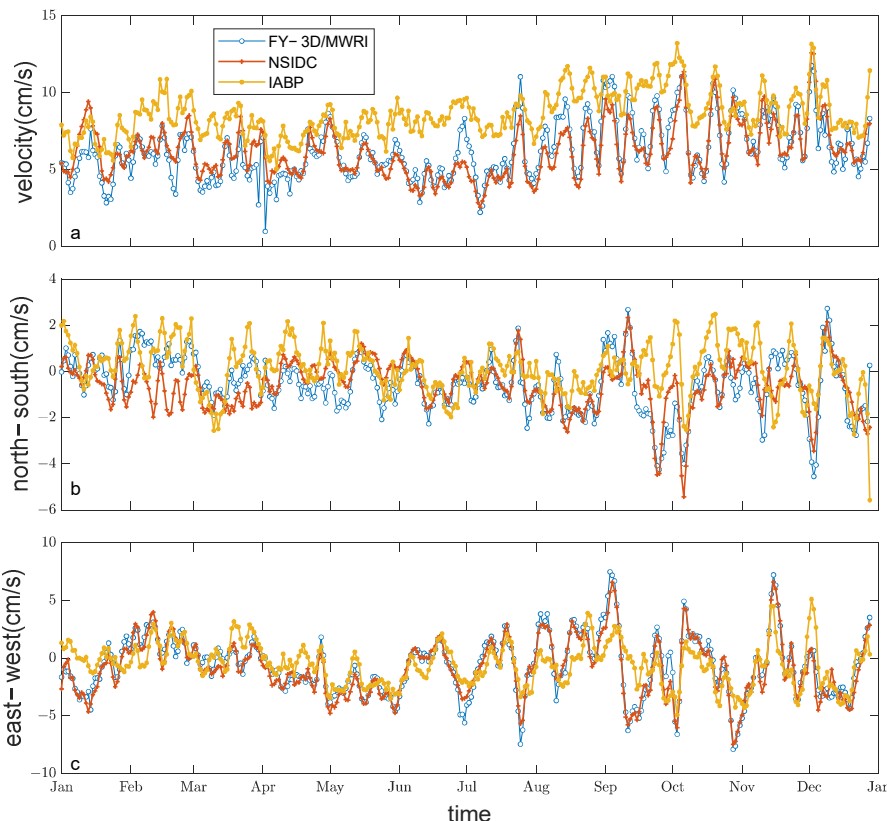

**Figure 14.** Comparison of the daily average SIM from the NSIDC, FY-3D/MWRI, and IABP buoy in 2019. (**a**) is the comparison of the absolute values of the SIM velocities, (**b**) is the comparison results of the north–south SIM velocity, (**c**) is the comparison results of the east–west SIM velocity.

We further performed an error analysis on the SIM daily averages data in 2019. Compared to the NSIDC SIM, the MAE and RMSE of the retrieved SIM from the FY-3D/MWRI $T_b$ data were 0.56 cm/s and 0.74 cm/s in the east–west direction and 0.72 cm/s and 0.93 cm/s in the north–south direction, respectively. Meanwhile, compared to the IABP buoy motion data, the MAE and RMSE of the NSIDC daily average SIM products were 1.45 cm/s and 1.82 cm/s in the east–west direction and 1.08 cm/s and 1.35 cm/s in the north–south direction, and the MAE and RMSE of the daily average SIM results obtained in this study were 1.50 cm/s and 1.90 cm/s in the east–west direction and 1.11 cm/s and 1.40 cm/s in the north–south direction, respectively.

The data of the ITP buoys placed by WHOI were also used to validate the SIM obtained in this study. The ITP buoys numbered 94, 102, 103, 104, 105, 107, 110, 111, 113, 114, and 116 were used in this study, as they could cover the entire 2019 year. Matching the ITP buoys to the SIM retrieved from the FY-3D/MWRI $T_b$ data, we obtained 1410 matching points in 2019, of which 319 were in summer and 1091 in winter.

Table 7 shows the errors of the SIM retrieved from the FY-3D/MWRI $T_b$ data and the NSIDC SIM product compared to the ITP buoy data in summer and winter 2019.

**Table 7.** Errors of retrieved SIM from $T_b$ data and the SIM of the NSIDC product compared to the ITP buoy data.

|  | FY-3D/MWRI | | NSIDC | |
|---|---|---|---|---|
|  | East–West $\delta/\sigma$ (cm/s) | North–South $\delta/\sigma$ (cm/s) | East–West $\delta/\sigma$ (cm/s) | North–South $\delta/\sigma$ (cm/s) |
| Summer | 1.40/2.20 | 1.33/2.28 | 0.76/1.37 | 0.91/1.76 |
| Winter | 0.89/1.49 | 0.83/1.28 | 0.70/1.00 | 0.71/0.98 |
| 2019 | 1.01/1.68 | 0.95/1.56 | 0.71/1.09 | 0.76/1.20 |

Figures 15–17 are the distribution of the matching points between the ITP data and SIM retrieved from the FY-3D/MWRI $T_b$ data and NSIDC SIM product. The matching points are concentrated around the red line, indicating that the NSIDC SIM and the SIM retrieved from the MWRI $T_b$ data show a good agreement with the ITP buoy motion data.

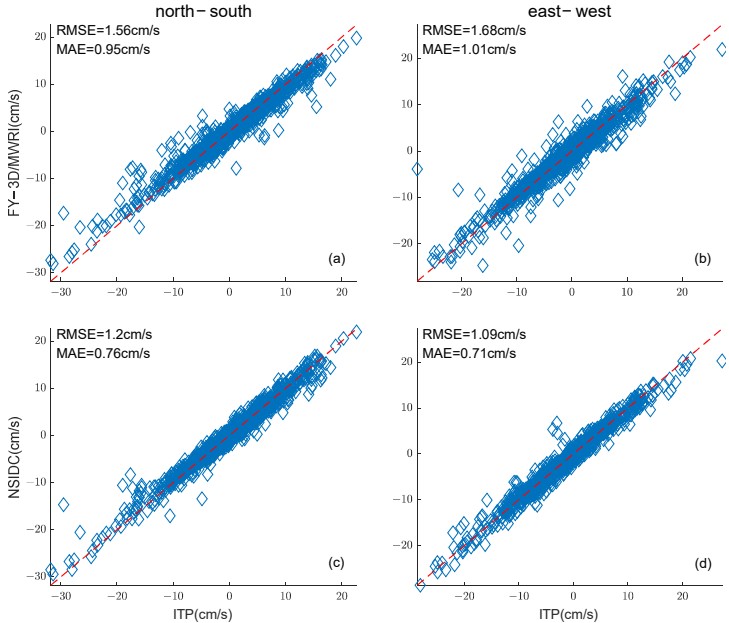

**Figure 15.** Distribution of the matching points in 2019. (**a**,**b**) are the distribution of the matching points between the ITP data and SIM retrieved from the FY-3D/MWRI $T_b$ data, (**c**,**d**) are the distribution of the matching points between the ITP data and NSIDC SIM product, in north–south and east–west direction, respectively.

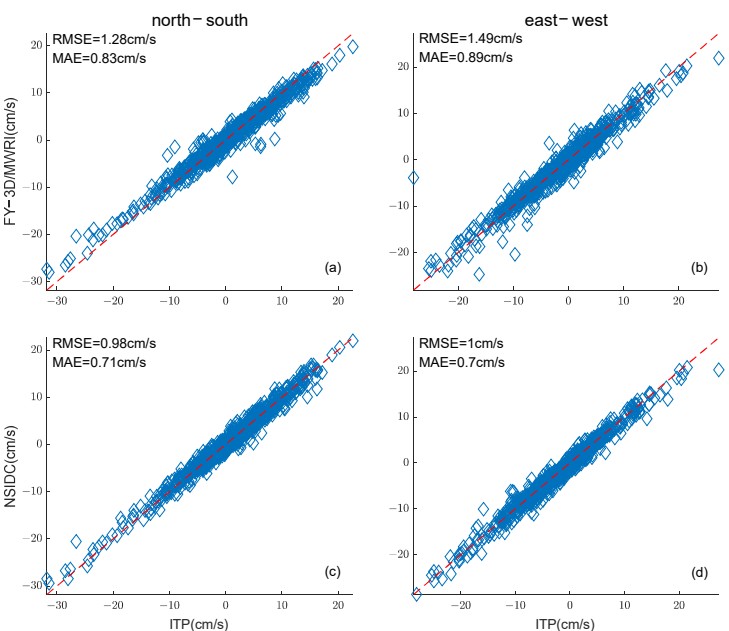

**Figure 16.** Distribution of the matching points in winter 2019. (**a**,**b**) are the distribution of the matching points between the ITP data and SIM retrieved from the FY-3D/MWRI $T_b$ data, (**c**,**d**) are the distribution of the matching points between the ITP data and NSIDC SIM product, in north–south and east–west direction, respectively.

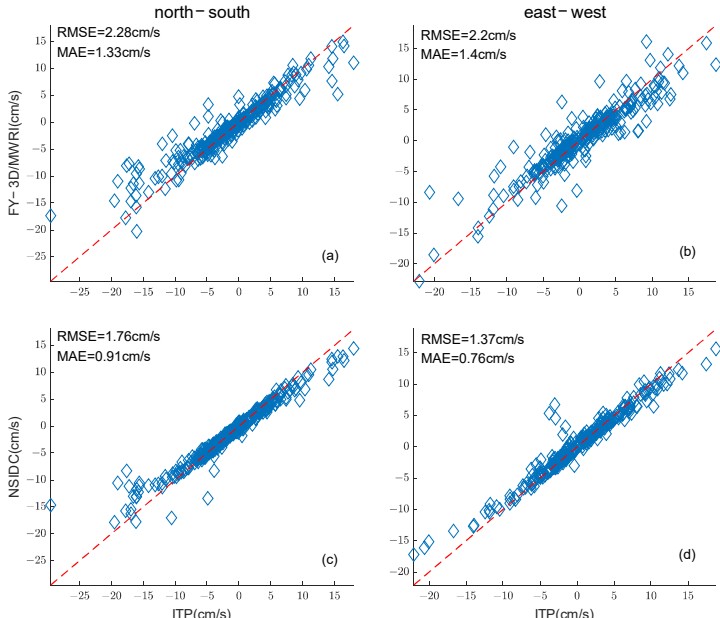

**Figure 17.** Distribution of the matching points in summer 2019. (**a**,**b**) are the distribution of the matching points between the ITP data and SIM retrieved from the FY-3D/MWRI $T_b$ data, (**c**,**d**) are the distribution of the matching points between the ITP data and NSIDC SIM product, in north–south and east–west direction, respectively.

In Table 7, compared to the ITP buoy, the MAE and RMSE of the SIM indicate that the errors of the SIM retrieved in our study and the NSIDC SIM product were very close. The SIM retrieved from the MWRI $T_b$ data had a slightly larger error compared to the NSIDC SIM product.

From Table 7, we can learn that the RMSE and MAE of the 2019 SIM retrieved from the FY-3D/MWRI $T_b$ data were larger than the NSIDC SIM product in the east–west direction

by 0.59 cm/s and 0.30 cm/s and in the north–south direction by 0.36 cm/s and 0.19 cm/s, respectively. In summer, the RMSE and MAE of the SIM retrieved from the $T_b$ data were larger than the NSIDC SIM product in the east–west direction by 0.83 cm/s and 0.64 cm/s and in the north–south direction by 0.52 cm/s and 0.42 cm/s, respectively. In winter, the RMSE and MAE of the SIM retrieved from the $T_b$ data were larger than the NSIDC SIM product in the east–west direction by 0.49 cm/s and 0.19 cm/s, respectively, and in the north–south direction by 0.30 cm/s and 0.12 cm/s, respectively.

In Table 8, we further calculated the average percentage error (APE) of the SIM inversion results in this paper and the SIM of the NSIDC product compared to the ITP buoy data. Taking into account the number of valid matching data points, corresponding to Table 7, we conducted the average percentage error (APE) analysis in terms of summer, winter, and full year 2019. As can be seen from the results in Table 8, the calculated APE in winter is better than that in summer compared to the ITP buoy. At the same time, the APE of the SIM obtained in this paper is basically the same as the APE of the NSIDC data products in the east–west and north–south directions in winter, while the APE of the NSIDC data in summer is slightly better than the APE obtained in this paper.

**Table 8.** The average percentage error (APE) of the SIM inversion results and the SIM of the NSIDC product compared to the ITP buoy data.

| | FY-3D/MWRI | | NSIDC | |
|---|---|---|---|---|
| **Time** | **East–West APE (%)** | **North–South APE (%)** | **East–West APE (%)** | **North–South APE (%)** |
| Summer | −12.67 | −13.94 | −5.20 | −7.98 |
| Winter | −4.64 | −6.13 | −3.15 | −5.19 |
| 2019 | −6.54 | −7.92 | −3.62 | −5.81 |

## 5. Conclusions

In this paper, we presented a SIM retrieval method for the Arctic region based on the FY-3D/MWRI 89 GHz and 36.5 GHz $T_b$ data. In this study, the classical MCC method was improved in the following aspects: the search area was determined based on IABP buoy motion data, different thresholds were set for different Arctic regions, and the SIM results were oversampled in order to reduce quantification errors. The multisource data merging method was also applied to merge the SIM retrieved from $T_b$ data, NCEP/NCAR reanalysis wind data, and IABP buoy motion data.

Compared to the ITP buoy data, the RMSE and MAE of the 2019 SIM retrieved in this study were 1.68 cm/s and 1.01 cm/s in the east–west direction, and 1.56 cm/s and 0.95 cm/s in the north–south direction; the APE of SIM inversion results in 2019 reached −6.54% in the east–west direction and −7.92% in the north–south direction. Compared to the daily average of the NSIDC SIM product, the RMSE and MAE of the 2019 SIM retrieved in this study were 0.93 cm/s and 0.72 cm/s in the north–south direction and 0.74 cm/s and 0.56 cm/s in the east–west direction, respectively. A comparison of the monthly SIM from NSIDC and that from the MWRI $T_b$ data showed a good agreement between the two. As can be seen from the results, the modified MCC algorithm proposed in this paper based on the FY-3D/MWRI $T_b$ data is suitable for the retrieval of the Arctic SIM and the inversion results are highly accurate.

However, there are still some shortcomings in this study that should be improved in the future. We will improve the MCC method in the future so that it more accurately reflect sea ice conditions in different regions and seasons. Furthermore, the multisource data merging method also needs to be further studied.

**Author Contributions:** Conceptualization, H.C., K.N., J.L. and L.L.; methodology, H.C. and K.N.; formal analysis, K.N. and H.C.; investigation, K.N., H.C., J.L. and L.L.; data curation, K.N., H.C. and J.L.; writing—original draft preparation, H.C., K.N. and L.L.; writing—review and editing, H.C., K.N., L.L. and J.L. All authors have read and agreed to the published version of the manuscript.

**Funding:** This research was supported by the national key research and development project of China (2019YFA0607001).

**Data Availability Statement:** Data sharing not applicable.

**Acknowledgments:** We are grateful to the National Satellite Meteorological Center, NSIDC, NCEP/ NCAR, IABP, ITP, and POGOC for providing the research data.

**Conflicts of Interest:** The authors declare no conflict of interest.

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
