# Peer review of "Retrieval of Arctic Sea Ice Motion from FY-3D/MWRI Brightness Temperature Data"

_remotesensing, doi:10.3390/rs15174191_

Round 1

Reviewer 1 Report

Sea ice motion is an important topic now in Cryosphere science. Authors present a interesting study on sea ice motion estimation from Microwave data onboard FY-3D with an improved MCC method. Overall, the manuscript is well written in English, but I don't suggest to publish it in its current status due to some important issues not addressed in the manuscript as stated below.

1. Sea ice motion involves two basic quantities - speed and direction. Only discussing scalar speed is not enough for SIM. Authors should add more results and discussion about SIM direction in comparison to other IABP, ITP, and other independent SIM data products. 

2. Authors should introduce more about the MCC method, its advantages, disadvantages, uncertainties, and limitations in application. And the setting changes in MCC should not be deemed as method improvement, and the selection of  those threshold values should further address in terms of physical meaning and reason. 

3. Validation work should also be improved by adding the percentage errors (RMSE and MAE) in terms of reference data products. Small bias or MAE doesn't mean better if the change in percentage is larger. Sea ice usually moves slow, so even few centimeter change in a second is large regarding its "true" value. As said, no validation work for sea ice motion direction is done for this work that should be added in revision.

4. Since 36.5GHz Tb retrieved SIM is better than 89GHz, why did you merge both? 

5. Multi-source data merging method appears not to make much improvement on SIM as seen from Figures 8 through 14. 

6. Authors should give full name of SCM before using its abbreviation. 

Well written in English. 

Reviewer 2 Report

This paper describes the algorythm for retrieval of sea ice motion from satellite data aquired from Fengyun-3 satellties. The authors use maximal cross-correlation method with several improvements to calculate velocity distribution of sea ice in the Arctic Ocean in 2019. Then they compare the obtained results with NSIDC sea ice motion data and IABP buoy data. This comparison demonstrate high quality of the developed novel sea ice motion data.

The paper is scientifically sound, well written, and extends available data for Arctic sea ice studies. As a result, this paper could be published as is.

Author Response

We would like to thank the referee for reviewing the paper.

Reviewer 3 Report

Dear Authors,

The manuscript is well written and provides important results on SIM. I may suggest to add more references on SIM operationally used for ice services now:

Howell, S. E. L., Brady, M., and Komarov, A. S.: Generating large-scale sea ice motion from Sentinel-1 and the RADARSAT Constellation Mission using the Environment and Climate Change Canada automated sea ice tracking system, The Cryosphere, 16, 1125–1139, https://doi.org/10.5194/tc-16-1125-2022, 2022.

A. S. Komarov and D. G. Barber, "Sea Ice Motion Tracking From Sequential Dual-Polarization RADARSAT-2 Images," in IEEE Transactions on Geoscience and Remote Sensing, vol. 52, no. 1, pp. 121-136, Jan. 2014, doi: 10.1109/TGRS.2012.2236845.

Specific comments:

1) Some abbreviations (e.g. ITP) are not explained.

2) Please provide description of FY-3B/MWRI satellite instruments.

3) Lines 136-137: Please provide explanations why these different threshold values were chosen.

4) Can you please explain high values of errors in some areas shown in figures 6-13. These figures need more detailed legend and description in text.

Best regards,

Reviewer

Good quality of English Language.

Round 2

Reviewer 1 Report

The revised manuscript is much better than the original one, and all my concerns are addressed.